# Utilizing a Rapid Multi-Plug Filtration Cleanup Method for 72 Pesticide Residues in Grape Wines Followed by Detection with Gas Chromatography Tandem Mass Spectrometry

**DOI:** 10.3390/foods10112731

**Published:** 2021-11-08

**Authors:** Shaowen Liu, Aijuan Bai, Le Song, Nan Zou, Yongtao Han, Li Zhou, Chuanshan Yu, Changjun Li, Canping Pan

**Affiliations:** 1Department of Applied Chemistry, College of Science, China Agricultural University, Beijing 100193, China; Shaowenliu2016@163.com (S.L.); baiaijuan678@163.com (A.B.); songle0626@163.com (L.S.); zounan5211314@163.com (N.Z.); ythan@rcees.ac.cn (Y.H.); chuanshan.yu@gmail.com (C.Y.); 2Hunan Institute of Agricultural Environment and Ecology, Changsha 410125, China; ricroot@sina.com; 3Tea Research Institute, Chinese Academy of Agricultural Sciences, Hangzhou 310008, China; lizhou@tricaas.com

**Keywords:** pesticide residue, GC–MS/MS, m-PFC, wine

## Abstract

A convenient and fast multi-residue method for the efficient identification and quantification of 72 pesticides belonging to different chemical classes in red and white grape wines has been developed. The analysis was based on gas chromatography tandem quadrupole mass spectrometric determination (GC–MS/MS). The optimization strategy involved the selection of the amount of multi-walled carbon nanotubes (MWCNTs) and the number of cleanup procedure cycles for multi-plug filtration cleanup (m-PFC) to achieve ideal recoveries and reduce the sample matrix compounds in the final extracts. The optimized procedure obtained consistent recoveries between 70.2 and 108.8% (70.2 and 108.8% for white wine, and 72.3 and 108.4% for red wine), with relative standard deviations (RSDs) that were generally lower than 9.2% at the three spiking levels of 0.01, 0.05 and 0.1 mg/kg. The linearity was studied in the range between 0.002 and 0.1 mg/kg using pesticide standards prepared both in pure solvent and in the presence of the matrix, showing coefficients of determination (R^2^) higher than 0.9495 for all the pesticides. To improve accuracy, matrix-matched calibration curves were used for calculating the quantification results. Finally, the method was used successfully for detecting pesticide residues in commercial grape wines.

## 1. Introduction

Grape wine is one of the most commonly consumed alcoholic beverages in the world. In addition to its distinctive flavor, the moderate consumption of wine is correlated with reduced risks of both mortality and morbidity from human cardiovascular disease [1] and oxidative damage [2]. The worldwide consumption of wine is increasing steadily and has reached up to 240 Mhl per year, according to records collected by the International Organisation of Vine and Wine over recent years [3].

During the grape cultivation period, it is common practice in vineyards to use pesticides, such as fungicides, insecticides and herbicides, to obtain high production. Grapes receive multiple doses of pesticides, which may partly transfer into wine [4,5,6,7,8]. In previous market surveillance studies [9,10,11,12,13], metalaxyl, procymidone, fenhexamid, cyprodinil, azoxystrobin and iprodione were detected in commercial grape wines; tebuconazol, metalaxyl and cyprodinil represent the most frequently detected pesticides. The risk of residues from these pesticides being present in wines implies a health hazard. For this reason, there are maximum residue limits (MRLs) set by current legislation [14]. To date, and with regard to grapevine products, MRLs have only been set for grapes, taking the transfer in the wine into account. The MRLs for wine are still not widely established [15,16,17]. Therefore, it is important to develop simple, rapid, environmentally friendly and sensitive analytical methods for the determination of trace levels of pesticide residues in wine samples in order to evaluate their safety and possible risk to human health.

Sample preparation is considered one of the most important steps in any procedure of pesticide residue analysis. The analysis of pesticide residues in wine is challenging due to the complexity of the matrix, which contains alcohol, organic acids, sugars and polyphenols (e.g., anthocyanins, flavonols and tannins). Many effective preparation methods of wine samples have been reported, including liquid–liquid extraction (LLE) with different organic solvents [11,18,19,20], solid-phase extraction (SPE) with reversed-phase C18 or polymeric sorbents [21,22,23,24,25,26,27], solid-phase microextraction (SPME) [22,28,29] and ultrasound-assisted emulsification microextraction (USAEME) [28,30], single drop liquid–liquid microextraction (LLME) [31,32,33,34], membrane-assisted solvent extraction [4] and dispersive liquid–liquid microextraction (DLLME) [35,36,37].

The QuEChERS (Quick, Easy, Cheap, Effective, Rugged and Safe) method is a sample preparation technique that was first reported in 2003 by Anastassiades et al. [38]. The QuEChERS cleanup technique belongs to the dispersive solid-phase extraction (dispersive SPE) class [39]. To date, there have been many reports on the application of QuEChERS-based methods for analyzing pesticides in wines [7,17,40,41,42,43,44,45,46,47,48,49].

Carbon nanotubes (CNTs) are novel and interesting carbonaceous materials first reported by Iijiama in 1991 [50]. These are classified into single-walled carbon nanotubes (SWCNTs) and multi-walled carbon nanotubes (MWCNTs) on the basis of the carbon atom layers in the walls of the nanotubes [51,52]. In recent years, MWCNTs have been reported to be used as effective SPE materials in the extraction of pesticides [53,54,55,56].

In our previous study, MWCNTs were used as alternative reversed-dispersive solid phase extraction materials in the multi-residue analysis of pesticides via the QuEChERS method [57,58,59]. They were mixed with other sorbents such as PSA, GCB and C18 for the dispersive cleanup of acetonitrile extracts from complex samples such as tea [58], scallions, ginger and garlic [60,61]. The new multi-plug filtration cleanup (m-PFC, Figure 1) procedure developed by our group, the solid-phase sorbents which were constituted with MWCNTs, other sorbents and anhydrous magnesium sulfate were packed in a short syringe cartridge. The syringe needle was kept under the surface of the extract, and the syringe piston was pushed and pulled for several cycles in order to adsorb the interfering substances and to remove water. The m-PFC method was very rapid, taking about 10 s to perform without any solvent evaporation [59,62,63,64].

The main objective of this work was to develop a fast, sensitive and reliable analytical m-PFC method. In this work, 72 pesticides with different chemical structures in wine were determined using GC-MS/MS. This method was successfully applied for market survey samples.

## 2. Materials and Methods

### 2.1. Chemicals and Materials

The initial sample preparation was identical to that used for the QuEChERS method [38]. Standard compounds were provided by the Institute of the Control of Agrochemicals, Ministry of Agriculture, China. The purity of the standard pesticides was 95–99%. Stock solutions of 10 mg/L of the pesticide mixture were prepared in acetonitrile and stored at −20 °C. The working solutions were prepared daily. HPLC-grade acetonitrile was obtained from Fisher Chemicals (Fair Lawn, NJ, USA). Analytical-reagent-grade anhydrous sodium chloride (NaCl) and magnesium sulfate (MgSO4) were obtained from Sinopharm Chemical Reagent (Beijing, China). Tianjin Bonna-Agela Technologies (Tianjin, China) provided MWCNTs with different average external diameters and PSA. The MWCNTs were dried for 2 h at 120 °C to remove the absorbed water and then kept in desiccators for storage.

### 2.2. Apparatus and Conditions

Centrifugation was performed with two different instruments: an Anke TDL-40B centrifuge equipped with a bucket rotor (4 × 100 mL) (Shanghai, China) and a SIGMA 3K15 microcentrifuge equipped with angular rotor (24 × 2.0 mL) (Sigma Laborzentrifugen GmbH, Osterode am Harz, Germany). A QL-901 Vortex (Kylin-bell Lab Instruments, Jiangsu, China) was used for preparing the samples. A Meiling BCD-245W refrigerator freezer (Beijing, China) was used to control the temperature of the samples.

Determinations were performed using an Agilent 7000 A triple-quadrupole mass spectrometer interfaced to an Agilent 7890 A GC. An Agilent Technologies analytical capillary column (HP-5MS; 30 m × 250 μm × 0.25 μm film thickness) was used for GC separation, with helium (99.9999%) as the carrier gas at a constant flow rate of 1.2 mL/min. The column temperature was initially set at 50 °C and held for 1 min, then increased to 130 °C (and held for 1 min) at a rate of 30 °C/min, then increased to 250 °C at a rate of 5 °C/min, and finally increased to 290 °C at a rate of 10 °C/min, which was held for 5 min. The temperature of the injector port was 250 °C, and a volume of 1 μL was injected in splitless mode. The total running time was 38 min.

The mass spectrometer was operated in electron ionization mode (70 eV). The default instrument settings of a collision gas flow for N_2_ of 1.5 mL/min and He of 2.25 mL/min, and a quadrupole temperature of 150 °C, were used in all the MS/MS experiments. The detector voltage was automatically set by the instrument after automated MS/MS tuning, which was typically 1250 V. A full autotune of the mass spectrometer, using the default parameters of the instrument, was performed before each sequence. Agilent MassHunter was used for instrument control and data acquisition/processing. For the final multiple reaction monitoring (MRM) acquisition method, two ion transitions at the experimentally optimized collision energy (CE) were monitored for each analyte. Both pairs of the MRM transitions were used for confirmation analysis, and the most sensitive transitions were selected for quantification analysis to obtain better separation efficiency. Table 1 summarizes the optimized MS/MS conditions for the individual analytes and their typical retention times (RT).

### 2.3. Sample Preparation

A QuEChERS-based approach was adapted to isolate the 72 analytes in the wine samples. The samples were obtained from a local supermarket and homogenized with a blender for 1 min at room temperature. For the determination of the recovery, the homogenized samples (10.0 ± 0.1 g) were spiked by adding the standard stock solutions at three concentrations: 0.01, 0.05 and 0.1 mg/kg. The spiked samples were set aside for 30 min before extraction.

An amount (10.0 ± 0.1 g) of each wine sample was weighed into a 50 mL centrifuge tube and 10 mL of acetonitrile was added. The resulting solution was shaken using a vortex for 1 min; then, 1 g of sodium chloride and 4 g of anhydrous magnesium sulfate were added. The tube was cooled immediately to room temperature in an ice-water bath. The centrifuge tube was shaken vigorously for 1 min to prevent salt agglomeration before centrifugation at 3800 rpm for 5 min. The 1 mL supernatant was used for further m-PFC.

### 2.4. m-PFC Procedures

The m-PFC procedure involved the following steps (shown in Figure 1): 1 mL of the supernatant was introduced into a 2.0 mL centrifuge tube. The sorbents (including 150 mg of anhydrous MgSO_4_) in the column were adopted from the optimized d-SPE sorbents. As shown in Figure 1, the syringe needle was kept under the surface of the extract; then, the syringe piston was pulled and pushed to let the extracts pass through the sorbents for the purpose of cleaning. Finally, the layer was filtered through a 0.22 μm filter membrane. The extract was placed into a GC vial for chromatographic analysis.

### 2.5. Method Performance

The analytical method was validated according to the following parameters: the linearity, limit of quantification (LOQ), limit of detection (LOD), precision and accuracy. The test of linearity used matrix-matched calibration by analyzing samples of red wine and white wine. The precision and accuracy experiments were carried out in five replicates, each at three fortification levels (0.01, 0.05 and 0.1 mg/kg) for the sample matrix. According to SANTE/12682/2019 [65], the LOQs were determined as the concentrations of analyte giving a signal-to-noise ratio (S/N) ≥ 3 and analyte peaks from both product ions in the extracted ion chromatograms must fully overlap. Ion ratio from sample extracts should be within ±30% (relative) of average of calibration standards from the same sequence.

### 2.6. Analysis of Grape Wine Samples

Grape wine samples, including fifty red grape wines and twenty white grape wines, were purchased at supermarkets in Beijing and they belonged to several vintages between 2011 and 2018. Seventy samples were produced in wineries from different countries: China (42), France (7), Spain (5), Portugal (5), Italy (4), Australia (4) and the USA (3). The alcoholic strength ranged from 10% to 15% (Alc/vol). Bottled wines were stored in their original packaging at 5 °C.

## 3. Results and Discussion

### 3.1. Amount of the MWCNTs

After the analytes had been extracted using 10 mL of acetonitrile, followed by the partitioning of the analyte molecules in an organic solvent in the presence of a salt mixture (the salting-out effect), 1 mL of the acetonitrile phase was further cleaned by the m-PFC procedure. Zhao et al. [62] found that different amounts of MWCNT sorbents had a significant influence on the purification and recovery of the pesticide extracts. To evaluate the effect of this parameter, different amounts of MWCNT were investigated in the same procedure. The amount of sorbent material was progressively increased from 5 mg to 10, 15 and 20 mg. The experiment was performed using 1 mL of the acetonitrile extract at a spiking level of 0.1 mg/kg and it was then cleaned by the m-PFC method with different amounts of MWCNT. The recovery of most of the analytes increased with the amount of MWCNTs and the results were within an acceptable range: 70–120% for red wine. As shown in Figure 2, upon increasing the amount of MWCNTs from 5 to 10 mg, the recovery levels for epoxiconazole, profenofos, azoxystrobin and bifenthrin remained acceptable (70–103%). However, the recovery decreased to 33–69% when the amount of MWCNTs was increased to 15 and 20 mg. In addition, although better recovery was achieved with 5 mg of MWCNT materials, the performance was not as good as that with 10 mg, and there was more chromatography interference when 5 mg was used. The recovery was also acceptable with 10 mg of MWCNTs. Consequently, 10 mg (1 mL of the extract) was used as the optimum amount for m-PFC in further studies, since acceptable recovery and good cleanup performance was obtained with this amount.

### 3.2. Optimization of the m-PFC Procedure Cycle Times

In order to obtain the best recovery and cleanup performance, the cycles of pulling and pushing during the m-PFC procedure were optimized. The recovery was acceptable with one and two cycles of pulling and pushing, but the cleanup performance was not as good as that with three cycles and there was more chromatography interference for one or two pull–push cycles. In addition, four cycles were also tested, but there was no significant difference in cleanup performance from that with three cycles. As a result, three cycles of pulling and pushing were chosen for the optimized m-PFC procedure. Figure 3 shows the purification effects of different cleanup cycles.

### 3.3. Validation of the Method

#### 3.3.1. Linearity and Matrix Effects

Linearity was studied in the range of 0.002–0.1 mg/L for all the pesticides at five calibration levels (0.002, 0.005, 0.01, 0.02, 0.05 and 0.1 mg/L) by a matrix-matched standard calibration in blank extracts of red wine and white wine. Linear calibration graphs were constructed by plotting analyte concentrations versus the relative peak areas of the calibration standards. The linearity values, calculated as the determination coefficients (R^2^) for each pesticide from the matrix-matched calibration (m-PFC cleanup) plots, are shown in Table 2. The quantitative results of the detection method greatly depend on its calibration. Both pure solvent-based as well as matrix-matched calibrations gave R^2^ values better than 0.985. This was remarkable, considering the complexity of the matrices. The matrix effects (ME) were evaluated in terms of slope ratios: 100 × (1-slope acetonitrile/slope matrix) [62,66].

The matrix effects include enhancement or suppression effects, so the concentration results obtained can be erroneous, depending on the solvent calibration curves [18]. To examine the matrix effects, matrix-matched standards were compared with solvent standards. Table 2 summarizes the results. In our work, it was considered that, if the value was in the range of −10 to 10%, the matrix effect could be ignored; if the value was lower than −10% or higher than 10%, this showed a matrix-suppression or an enhancement effect, respectively [62,66]. The results show that, in red and white wine, 45 and 25 of the pesticides presented an enhancement effect (ME > 0), respectively, and the other 27 and 47 of the pesticides showed a suppression effect (ME < 0), respectively; 44 and 34 of the pesticides expressed distinct matrix-suppression and enhancement effects. Therefore, for more accurate results, validation experiments were performed for pesticide residue concentrations in non-compliant samples, calculated using matrix-matched calibration standards and excluding any influence produced by matrix effects, as recommended in SANTE/12682/2019 [65]. In order to overcome the adverse impact of matrix effects on the quantified results, we calibrated the sample results with matrix-matched standards to guarantee the correct quantification of the pesticide concentrations in real samples.

#### 3.3.2. Recovery and Precision

The recovery and repeatability of the method were established to evaluate the method’s performance. The repeatability and the accuracy of the method were tested by carrying out five consecutive extractions (*n* = 5) of spiked matrices at three concentration levels (0.01, 0.05 and 0.1 mg/kg). All the recovery values were determined from analyses of the 72 pesticides in the matrices. The values were calculated using matrix-matched calibration standards, as stated in Section 3.3.1. Table 3 shows detailed recovery and repeatability data for all the pesticides analyzed in wine matrices. The recovery rates of all the pesticides were in the range of 70.2–108.8% (between 70.2 and 108.8% for white wine, and between 72.3 and 106.0% for red wine). The relative standard deviations (RSDs) were below 8.3% for all the cases. All the recovery values and RSDs are in the acceptable range of SANTE/12682/2019 [65].

#### 3.3.3. Limits of Quantitation and Limits of Detection

The described method was tested for the simultaneous extraction and determination of 72 analytes in wine matrices, which manifested varying LODs and LOQs. Since LODs and LOQs are matrix-dependent, it is recommended to perform matrix-matched calibrations for the quantitative analysis of unknown samples in complex matrices. Table 3 shows the LOD and LOQ values for the pesticides in wine under study. The LODs and LOQs ranged from 0.002 to 0.01 mg/kg and from 0.01 to 0.05 mg/kg, respectively.

In general, the validation data for all the analytes were in accordance with the EU guidelines (2019) SANTE/12682/2019 [65] for pesticide residue analysis, reflecting the good performance of the method. Comparison with other QuEChERS methods for determining pesticides in grape wines, the m-PFC method which our group has proposed, showed significant advantages in terms of the amount of time that is consumed, each sample took less than two minutes for m-PFC cleanup, so it would be time-saving during the processing of enormous samples. Regarding recoveries, RSDs, LOQs and the number of pesticides, the m-PFC method presented similar effects when compared to other methods (Table 4).

### 3.4. Method Application

The developed QuEChERS method with a m-PFC cleanup step was applied to real samples. Seventy reference samples (50 for red wine samples and 20 for white wine samples) from the supermarkets in Beijing were treated and analyzed by GC-MS/MS. Since MRLs have not yet been set for wine, according to SANTE/2020/12830 [66] and EC 657/2002 [67], it is accepted that the MRLs for wine are the same as those for wine grapes. Table 5 shows the detected concentrations of pesticides in the real samples from supermarkets in Beijing. Pesticide residues were detected in seven samples (10%). The most frequently detected pesticides were difenoconazole (2.9%), pyridaben (4.3%), carbosulfan (2.9%), pyr imethanil (1.4%), propyzamide (1.4%), simazine (4.3%) and atrazine (4.3%). To date, and with regard to grapevine products, MRLs have only been set for grapes, taking the transfer in the wine into account. The MRLs for wine are still not widely established [15,16,17]. Therefore, we referred to the maximum permitted residue levels set by EU2018/555 [68], The concentrations of pesticides in the selected wine samples did not exceed the permitted residue levels.

## 4. Conclusions

An efficient and effective m-PFC multi-residue method was developed for the determination of 72 pesticides in wine by GC–MS/MS. The m-PFC method, which could be carried out without any solvent evaporation, vortexing or centrifugation procedure, proved to be a simple and rapid cleanup method. The method achieved high-quality results (good repeatability and recovery, and a wide analytical scope) and had several practical benefits (low cost, little labor, high sample throughput, hardly any waste and low labware equipment and space demands). The method was found to be very sensitive and gave a LOQ of <0.05 mg/kg for all the analytes. In conclusion, m-PFC could be used as a feasible, convenient and rapid high-throughput cleanup method for the analysis of analytes in wine.

## Figures and Tables

**Figure 1 foods-10-02731-f001:**
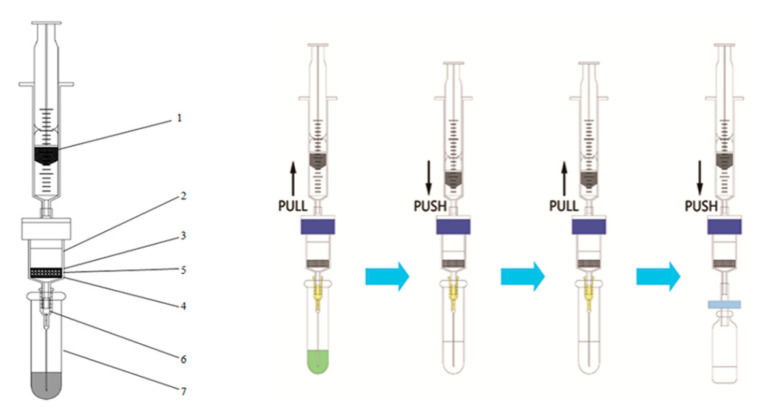
Schematic diagram of an m-PFC setup (original drawing from Zhao et al. [62]): 1, syringe; 2, column; 3, PE filter (**upper**); 4, PE filter (**lower**); 5, MWCMNs (10 mg) and anhydrous magnesium sulfate (150 mg); 6, syringe needle; 7, 2.0 mL microcentrifuge tube.

**Figure 2 foods-10-02731-f002:**
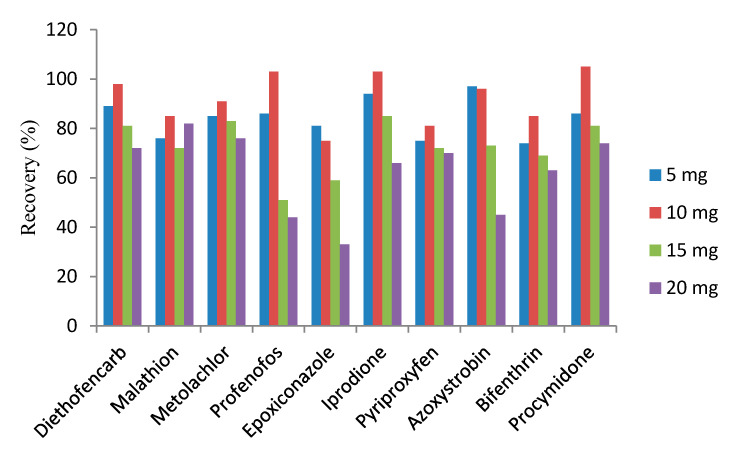
Effects of the amount of MWCNTs on recovery.

**Figure 3 foods-10-02731-f003:**
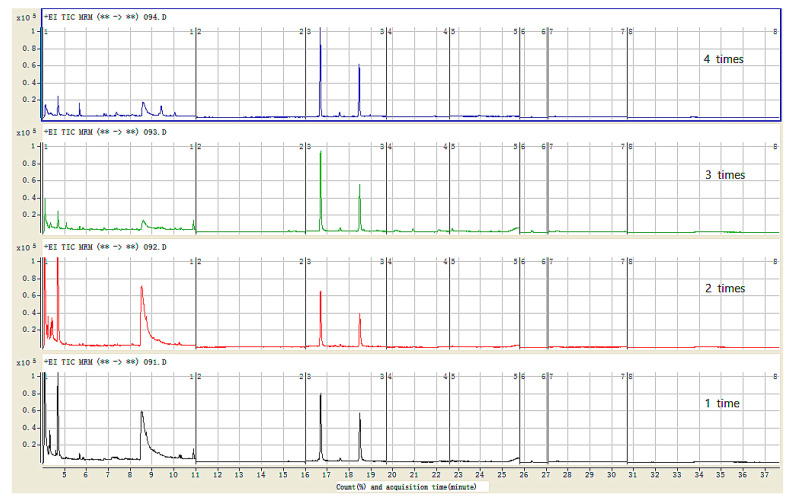
The optimization of m-PFC procedure cycle times by pulling and pushing to red wine blank samples (the initial value for each cleanup sample: 2 mL).

**Table 1 foods-10-02731-t001:** MRM parameters of 72 pesticides in wine determined by GC-MS/MS.

No.	Pesticide	RT (min)	Quantification Transition ^a^	Confirmation Transition ^a^
1	dichlorvos	5.22	109→79 (5)	185→93 (10)
2	*o*-phenylphenol	10.21	170→169 (10)	169→141 (10)
3	sulfotep-ethyl	13.31	322→146 (25)	322→65 (40)
4	phorate	13.41	121→65 (10)	260→75 (5)
5	simazine	14.32	201→172 (10)	186→68 (25)
6	thiabendazole	14.33	201→130 (25)	201→174 (15)
7	carbofuran	14.51	164→149 (10)	164→131 (20)
8	indoxacarb	14.51	218→203 (15)	264→176 (15)
9	atrazine	14.52	171.9→69 (15)	172→43 (30)
10	acephate	14.52	136→94 (10)	136→42 (10)
11	clomazone	14.60	124.9→89 (20)	204→107 (20)
12	terbufos	15.01	231→129 (25)	231→175 (10)
13	pyrimethanil	15.33	198→118 (25)	198→156 (25)
14	acetochlor	17.18	223→132 (20)	146→118 (10)
15	methyl parathion	17.23	263→109 (15)	263→79 (30)
16	dimethoate	17.36	125→79 (5)	125→93 (10)
17	tolclofos-methyl	17.36	265→250 (15)	265→93 (25)
18	iprovalicarb I	17.88	158→98 (10)	158→116 (10)
19	fenitrothion	18.29	277→260 (5)	277→109 (20)
20	ethofumesate	18.44	286→207 (5)	286→179 (15)
21	carbosulfan	18.74	160→104 (10)	160→57 (15)
22	malathion	18.76	173→99 (15)	173→117 (15)
23	metolachlor	18.87	162→133 (15)	162.2→132 (25)
24	fenthion	18.99	278→109 (10)	278→125 (15)
25	diethofencarb	19.01	267→225 (5)	196→168 (5)
26	chlorpyrifos	19.06	314→258 (15)	314→286 (15)
27	triadimefon	19.26	208→181 (10)	208→111 (15)
28	isocarbophos	19.34	136→108 (14)	230→212 (8)
29	cyprodinil	19.977	225→224 (10)	224→208 (20)
30	metazachlor	20.18	209→132 (20)	133→117 (25)
31	pendimethalin	20.25	252→162 (10)	252→161 (20)
32	chlorfenvinphos	20.68	267→159 (20)	267→81 (40)
33	fipronil	20.78	367→213 (30)	367→228 (30)
34	procymidone	20.90	283→96 (10)	283→255 (10)
35	vinclozolin	20.90	212→145 (15)	212→172 (25)
36	methidathion	21.18	145→85 (5)	145→58 (15)
37	butachlor	21.76	237→160 (5)	188.1→160 (10)
38	flutriafol	21.94	164→109 (20)	219→123 (15)
39	carbaryl	22.04	144→116 (15)	144→114 (30)
40	napropamide	22.04	128→72 (10)	271→128 (5)
41	hexaconazole	22.14	213.9→172 (20)	214→159 (20)
42	profenofos	22.40	208→63 (35)	208→98 (25)
43	oxadiazon	22.72	175→112 (15)	302→175 (13)
44	iprovalicarb II	22.73	158→98 (10)	158→116 (10)
45	carboxin	22.76	235→143 (5)	144→87 (5)
46	oxyfluorfen	22.97	252→252 (5)	252→196 (20)
47	flusilazole	23.05	233→152 (20)	233→165 (20)
48	kresoxim-methyl	23.13	206→116 (5)	206→131 (10)
49	metalaxyl	23.13	206→132 (5)	206→162 (20)
50	diniconazole	24.11	268→232 (15)	270→234 (15)
51	triazophos	24.72	161→134 (5)	257→162 (5)
52	propiconazole Ⅰ	25.27	259→173 (15)	261→175 (15)
53	propiconazole Ⅱ	25.46	259→69 (12)	259→191 (5)
54	propyzamide	25.47	173→145 (20)	175→147 (20)
55	diclofop-methyl	25.96	253→162 (15)	340→253 (15)
56	epoxiconazole	26.55	192→138 (10)	192→157 (5)
57	iprodione	26.85	314→245 (10)	314→271 (20)
58	cypermethrin-Ⅰ	27.33	181→152 (30)	181→127 (35)
59	bifenthrin	27.33	181→165 (25)	181→166 (25)
60	bifenox	27.77	311→279 (10)	311→216 (20)
61	pyriproxyfen	28.61	136→78 (25)	-
62	cypermethrin Ⅱ	28.91	181→152 (30)	181→127 (35)
63	beta-cypermethrin	28.92	181→152 (30)	181→127 (35)
64	cypermethrin Ⅲ	29.26	181→152 (30)	181→127 (35)
65	permethrin Ⅰ	30.58	183→153 (20)	183→168 (20)
66	pyridaben	30.52	147→117 (20)	147→132 (10)
67	permethrin Ⅱ	30.37	183→115 (25)	183→77 (30)
68	cypermethrin Ⅳ	30.58	181→152 (30)	181→127 (35)
69	difenoconazole	33.61	323→265 (10)	265→139 (25)
70	azoxystrobin	34.40	344→329 (15)	253→172 (20)
71	deltamethrin Ⅰ	33.62	181→152 (25)	253→172 (10)
72	deltamethrin Ⅱ	33.92	181→152 (25)	253→172 (10)

^a^ Collision energy (eV) is given in parentheses.

**Table 2 foods-10-02731-t002:** Linearity parameters (range, slope and R^2^) obtained by using standards in acetonitrile and by matching, as well as matrix effects measured as 100 × (1-slope acetonitrile/slope matrix).

Pesticide	Linearity Range (mg/kg)	Acetonitrile		Red Wine			White Wine		
		Slope	R^2^	Slope	R^2^	Matrix Effect, %	Slope	R^2^	Matrix Effect, %
dichlorvos	0.002–0.1	9.6 × 10^4^	0.9924	1.1 × 10^5^	0.9977	9.4	9.5 × 10^4^	0.9999	−1.7
*o*-phenylphenol	0.002–0.1	6.8 × 10^5^	0.9989	8.5 × 10^5^	0.9912	19.5	7.0 × 10^5^	0.9958	2.2
sulfotep-ethyl	0.002–0.1	3.3 × 10^5^	0.9916	3.6 × 10^5^	0.9982	8.5	3.4 × 10^5^	0.9920	3.6
phorate	0.002–0.1	2.2 × 10^5^	0.9897	2.3 × 10^5^	0.9899	6.1	2.3 × 10^5^	0.9957	5.9
simazine	0.002–0.1	3.4 × 10^4^	0.9878	3.9 × 10^4^	0.9985	12.5	3.1 × 10^4^	0.9990	−7.4
thiabendazole	0.005–0.1	6.8 × 10^3^	0.9855	5.9 × 10^3^	0.9959	−14.9	6.6 × 10^3^	0.9956	−3.8
carbofuran	0.002–0.1	3.1 × 10^5^	0.9968	3.3 × 10^5^	0.9986	8.1	3.2 × 10^5^	0.9921	4.5
indoxacarb	0.005–0.1	3.0 × 10^3^	0.9997	2.8 × 10^3^	0.9922	−5.9	2.5 × 10^3^	0.9894	−17.0
atrazine	0.002–0.1	1.7 × 10^4^	0.9989	2.0 × 10^4^	0.9954	11.1	1.5 × 10^4^	0.9853	−15.3
acephate	0.005–0.1	3.3 × 10^3^	0.9991	3.8 × 10^3^	0.9858	13.6	3.4 × 10^3^	0.9887	2.6
clomazone	0.002–0.1	3.1 × 10^5^	0.9926	3.7 × 10^5^	0.9495	14.8	3.2 × 10^5^	0.9885	3.2
terbufos	0.002–0.1	2.1 × 10^4^	0.9917	2.5 × 10^4^	0.9916	17.4	2.5 × 10^4^	0.9977	17.1
pyrimethanil	0.002–0.1	1.7 × 10^5^	0.9957	1.5 × 10^5^	0.9987	−10.1	1.6 × 10^5^	0.9986	−6.9
acetochlor	0.002–0.1	8.8 × 10^4^	0.9935	1.2 × 10^5^	0.9984	24.1	9.5 × 10^4^	0.9910	6.9
methyl parathion	0.002–0.1	3.8 × 10^5^	0.9996	4.1 × 10^5^	0.9941	7.2	3.7 × 10^5^	0.9915	−2.1
dimethoate	0.002–0.1	1.0 × 10^6^	0.9919	9.3 × 10^5^	0.9896	−7.2	8.9 × 10^5^	0.9994	−12.9
tolclofos-methyl	0.002–0.1	5.0 × 10^6^	0.9928	6.0 × 10^6^	0.9959	16.7	5.4 × 10^6^	0.9995	7.4
iprovalicarb I	0.002–0.1	5.5 × 10^3^	0.9937	6.8 × 10^3^	0.9866	18.7	4.9 × 10^3^	0.9956	−12.8
fenitrothion	0.002–0.1	8.4 × 10^4^	0.9930	9.5 × 10^4^	0.9859	12.3	8.5 × 10^4^	0.9943	1.9
ethofumesate	0.002–0.1	2.1 × 10^5^	0.9954	2.2 × 10^5^	0.9973	5.8	1.9 × 10^5^	0.9899	−11.2
carbosulfan	0.002–0.1	9.2 × 10^2^	0.9919	1.1 × 10^3^	0.9928	15.4	1.1 × 10^3^	0.9945	15.6
malathion	0.002–0.1	2.8 × 10^5^	0.9966	3.1 × 10^5^	0.9959	10.4	2.5 × 10^5^	0.9948	−11.9
metolachlor	0.002–0.1	4.5 × 10^5^	0.9987	5.5 × 10^5^	0.9866	18.2	4.5 × 10^5^	0.9942	−0.4
fenthion	0.002–0.1	1.4 × 10^5^	0.9934	1.6 × 10^5^	0.9967	7.5	1.4 × 10^5^	0.9984	−3.5
diethofencarb	0.002–0.1	3.7 × 10^5^	0.9997	4.0 × 10^5^	0.9819	6.5	3.9 × 10^5^	0.9894	5.6
chlorpyrifos	0.002–0.1	1.9 × 10^5^	0.9942	2.2 × 10^5^	0.9925	15.7	1.9 × 10^5^	0.9948	1.3
triadimefon	0.002–0.1	1.2 × 10^5^	0.9998	9.7 × 10^4^	0.9937	−24.1	1.1 × 10^5^	0.9951	−9.3
isocarbophos	0.002–0.1	3.3 × 10^5^	0.9929	3.5 × 10^5^	0.9958	5.8	3.4 × 10^5^	0.9935	4.2
cyprodinil	0.002–0.1	9.5 × 10^5^	0.9999	8.6 × 10^5^	0.9863	−10.4	7.7 × 10^5^	0.9864	−23.1
metazachlor	0.002–0.1	1.5 × 10^5^	0.9991	1.4 × 10^5^	0.9988	−9.2	1.8 × 10^5^	0.9987	14.0
pendimethalin	0.002–0.1	1.0 × 10^5^	0.9894	1.0 × 10^5^	0.9979	2.2	1.2 × 10^5^	0.9839	15.1
chlorfenvinphos	0.002–0.1	1.0 × 10^5^	0.9931	1.2 × 10^5^	0.9890	12.6	8.9 × 10^4^	0.9847	−14.9
fipronil	0.002–0.1	5.2 × 10^4^	0.9968	6.5 × 10^4^	0.9881	19.0	4.9 × 10^4^	0.9850	−6.6
procymidone	0.002–0.1	2.2 × 10^5^	0.9914	2.1 × 10^5^	0.9959	−3.2	1.8 × 10^5^	0.9854	−23.2
vinclozolin	0.002–0.1	6.6 × 10^3^	0.9934	6.0 × 10^3^	0.9948	−10.5	5.9 × 10^3^	0.9994	−13.2
methidathion	0.002–0.1	4.3 × 10^5^	0.9895	4.7 × 10^5^	0.9939	9.2	3.8 × 10^5^	0.9885	−11.3
butachlor	0.002–0.1	7.5 × 10^4^	0.9916	9.1 × 10^4^	0.9957	18.0	7.3 × 10^4^	0.9887	−1.7
flutriafol	0.002–0.1	1.3 × 10^5^	0.9967	1.4 × 10^5^	0.9955	10.7	1.2 × 10^5^	0.9916	−8.6
carbaryl	0.005–0.1	2.7 × 10^4^	0.9958	2.4 × 10^4^	0.9889	−13.9	2.6 × 10^4^	0.9943	−6.9
napropamide	0.002–0.1	1.7 × 10^5^	0.9938	1.7 × 10^5^	0.9998	1.5	1.6 × 10^5^	0.9979	−6.3
hexaconazole	0.002–0.1	5.1 × 10^4^	0.9996	4.6 × 10^4^	0.9964	−12.0	4.8 × 10^4^	0.9948	−6.7
profenofos	0.002–0.1	1.6 × 10^4^	0.9949	2.1 × 10^4^	0.9962	25.3	1.8 × 10^4^	0.9999	12.3
oxadiazon	0.002–0.1	3.0 × 10^5^	0.9925	3.0 × 10^5^	0.9969	−0.8	2.5 × 10^5^	0.9948	−18.9
iprovalicarb II	0.005–0.1	1.1 × 10^3^	0.9939	9.9 × 10^2^	0.9928	−5.7	9.3 × 10^2^	0.9972	−12.7
carboxin	0.002–0.1	2.5 × 10^5^	0.9916	3.4 × 10^5^	0.9896	28.6	2.4 × 10^5^	1.0000	−4.2
oxyfluorfen	0.002–0.1	2.3 × 10^5^	0.9984	2.6 × 10^5^	0.9927	11.5	2.0 × 10^5^	0.9942	−17.5
flusilazole	0.002–0.1	1.1 × 10^5^	0.9956	8.8 × 10^4^	0.9990	−24.3	9.5 × 10^4^	0.9940	−15.0
kresoxim-methyl	0.002–0.1	1.9 × 10^5^	0.9942	1.6 × 10^5^	0.9792	−19.6	1.8 × 10^5^	0.9998	−10.1
metalaxyl	0.002–0.1	1.2 × 10^5^	0.9967	9.7 × 10^4^	0.9963	−20.1	9.3 × 10^4^	0.9965	−25.2
diniconazole	0.002–0.1	1.5 × 10^5^	0.9924	1.3 × 10^5^	0.9928	−21.4	1.4 × 10^5^	0.9971	−14.1
triazophos	0.002–0.1	9.2 × 10^4^	0.9913	9.2 × 10^4^	0.9969	−0.8	9.1 × 10^4^	0.9857	−2.0
Propiconazole I	0.002–0.1	9.4 × 10^4^	0.9957	9.4 × 10^4^	0.9942	−0.8	9.1 × 10^4^	0.9920	−3.7
propiconazole II	0.002–0.1	2.1 × 10^5^	0.9962	2.2 × 10^5^	0.9967	2.5	1.9 × 10^5^	0.9854	−10.3
propyzamide	0.002–0.1	5.5 × 10^5^	0.9938	4.0 × 10^5^	0.9966	−37.3	4.9 × 10^5^	0.9801	−13.0
diclofop-methyl	0.002–0.1	1.1 × 10^5^	0.9989	1.3 × 10^5^	0.9894	18.3	1.0 × 10^5^	0.9900	−3.9
epoxiconazole	0.01–0.1	6.0 × 10^4^	0.9937	6.4 × 10^4^	0.9964	5.5	5.1 × 10^4^	0.9977	−17.2
iprodione	0.002–0.1	5.5 × 10^4^	0.9988	7.3 × 10^4^	0.9922	24.3	5.6 × 10^4^	0.9884	1.9
cypermethrin-I	0.002–0.1	2.2 × 10^4^	0.9936	2.4 × 10^4^	0.9960	8.4	2.1 × 10^4^	0.9978	−4.3
bifenthrin	0.002–0.1	8.8 × 10^5^	0.9942	1.0 × 10^6^	0.9920	12.3	8.8 × 10^5^	0.9920	0.4
bifenox	0.002–0.1	4.1 × 10^4^	0.9969	4.9 × 10^4^	0.9935	16.2	5.6 × 10^4^	0.9918	26.8
pyriproxyfen	0.002–0.1	2.0 × 10^5^	0.9960	2.6 × 10^5^	0.9919	24.4	2.0 × 10^5^	0.9912	3.3
cypermethrin II	0.005–0.1	5.3 × 10^4^	0.9938	4.7 × 10^4^	0.9933	−11.8	4.2 × 10^4^	0.9924	−25.4
beta-cypermethrin	0.005–0.1	3.4 × 10^4^	0.9962	3.7 × 10^4^	0.9957	9.9	3.2 × 10^4^	0.9909	−4.4
cypermethrin III	0.005–0.1	2.8 × 10^4^	0.9973	2.8 × 10^4^	0.9872	−1.7	2.5 × 10^4^	0.9955	−11.7
permethrin I	0.005–0.1	3.9 × 10^4^	0.9959	4.7 × 10^4^	0.9868	16.0	5.0 × 10^4^	0.9947	21.6
pyridaben	0.002–0.1	8.0 × 10^5^	0.9929	1.0 × 10^6^	0.9910	20.0	8.3 × 10^5^	0.9948	4.0
Permethrin II	0.005–0.1	2.5 × 10^4^	0.9954	2.3 × 10^4^	0.9925	−8.1	2.8 × 10^4^	0.9944	10.5
cypermethrin IV	0.005–0.1	7.7 × 10^3^	0.9966	9.2 × 10^3^	0.9890	16.2	9.3 × 10^3^	0.9978	17.0
difenoconazole	0.002–0.1	5.9 × 10^5^	0.9919	6.2 × 10^5^	0.9887	3.9	4.6 × 10^5^	0.9991	−28.4
azoxystrobin	0.005–0.1	2.0 × 10^4^	0.9961	2.4 × 10^4^	0.9921	16.2	1.6 × 10^4^	0.9958	−26.8
deltamethrin I	0.002–0.1	3.8 × 10^3^	0.9958	3.4 × 10^3^	0.9858	−13.1	3.6 × 10^3^	0.9957	−6.8
deltamethrin II	0.002–0.1	5.0 × 10^3^	0.9952	4.30 × 10^3^	0.9853	−16.1	4.20 × 10^3^	0.9875	−18.9

**Table 3 foods-10-02731-t003:** Average recovery, RSD, LOD and LOQ after application of the m-PFC procedure, determined by GC-MS/MS in wine.

Pesticide	Red Wine					White Wine				
	Recovery (RSD), %			LOD	LOQ	Recovery (RSD), %			LOD	LOQ
	0.01 mg/kg	0.05 mg/kg	0.1 mg/kg	(mg/kg)	(mg/kg)	0.01 mg/kg	0.05 mg/kg	0.1 mg/kg	(mg/kg)	(mg/kg)
dichlorvos	102.2 (6.2)	99.8 (3.2)	98.4 (8.1)	0.002	0.01	88.8 (2.1)	100.4 (8.0)	103.2 (2.5)	0.002	0.01
*o*-phenylphenol	100.5 (5.0)	88.4 (1.0)	83.4 (3.2)	0.002	0.01	81.8 (4.7)	97.1 (4.0)	86.9 (3.4)	0.002	0.01
sulfotep-ethyl	93.1 (1.4)	87.1 (3.1)	84.2 (2.7)	0.002	0.01	103.8 (4.8)	91.1 (1.1)	91.3 (3.1)	0.002	0.01
phorate	95.0 (4.9)	86.0 (2.1)	84.0 (2.5)	0.002	0.01	102.9 (4.1)	97.0 (1.2)	93.0 (0.5)	0.002	0.01
simazine	82.2 (4.4)	86.5 (1.8)	82.8 (1.3)	0.002	0.01	100.7 (2.6)	90.6 (4.4)	83.5 (2.7)	0.002	0.01
thiabendazole	84.7 (0.9)	99.3 (2.2)	88.1 (2.0)	0.002	0.01	-	91.3 (1.9)	86.9 (2.7)	0.01	0.05
carbofuran	97.3 (2.9)	89.1 (0.7)	92.5 (4.6)	0.002	0.01	108.4 (1.8)	100.7 (4.1)	100.2 (0.9)	0.002	0.01
indoxacarb	-	100.5 (2.3)	100.8 (5.9)	0.01	0.05	-	108.3 (0.8)	82.7 (2.1)	0.01	0.05
atrazine	94.9 (2.5)	86.8 (2.9)	84.8 (3.0)	0.002	0.01	96.0 (2.7)	95.9 (3.2)	91.3 (1.4)	0.002	0.01
acephate	-	87.6.0 (8.3)	94.5 (4.0)	0.01	0.05	-	90.0 (2.6)	95.3 (1.1)	0.01	0.05
clomazone	84.5 (5.6)	94.8 (2.5)	97.9 (5.8)	0.002	0.01	96.2 (4.0)	97.2 (3.4)	84.2 (1.1)	0.002	0.01
terbufos	73.7 (0.7)	83.6 (3.8)	79.4 (2.4)	0.002	0.01	92.4 (4.9)	98.9 (3.4)	89.4 (0.6)	0.002	0.01
pyrimethanil	99.7 (2.4)	92.7 (2.0)	96.1 (2.4)	0.002	0.01	94.8 (1.4)	90.2 (2.2)	85.1 (0.4)	0.002	0.01
acetochlor	87.1 (9.2)	92.7 (3.2)	84.1 (4.4)	0.002	0.01	105.3 (3.3)	101.6 (3.4)	90.8 (1.3)	0.002	0.01
methyl parathion	72.3 (2.8)	81.5 (2.6)	87.7 (6.8)	0.002	0.01	106.1 (3.0)	89.7 (4.4)	97.0 (1.7)	0.002	0.01
dimethoate	85.1 (3.0)	87.4 (2.9)	82.2 (2.1)	0.002	0.01	89.4 (2.3)	85.6 (2.0)	91.3 (5.9)	0.002	0.01
tolclofos-methyl	84.6 (3.4)	85.3 (2.4)	82.2 (2.4)	0.002	0.01	93.9 (3.8)	81.2 (3.5)	81.7 (2.0)	0.002	0.01
iprovalicarb I	-	94.2 (6.2)	104.3 (2.4)	0.02	0.05	99.3 (4.1)	96.3 (5.4)	84.2 (1.5)	0.002	0.01
fenitrothion	80.1 (3.2)	78.4 (3.3)	90.0 (4.4)	0.002	0.01	90.7 (3.7)	90.6 (5.7)	95.7 (1.7)	0.02	0.05
ethofumesate	94.7 (2.4)	89.4 (4.7)	90.7 (4.7)	0.002	0.01	96.1 (3.1)	100.2 (2.7)	93.4 (2.1)	0.003	0.01
carbosulfan	92.8 (2.7)	96.8 (3.2)	92.3 (3.0)	0.002	0.01	-	95.9 (4.8)	73.1 (5.5)	0.002	0.05
malathion	81.3 (3.9)	86.0 (4.0)	89.0 (6.7)	0.002	0.01	95.2 (4.3)	102.6 (0.3)	100.6 (2.4)	0.002	0.01
metolachlor	81.9 (5.0)	94.1 (3.1)	82.4 (5.2)	0.002	0.01	102.0 (1.6)	103.1 (3.8)	94.3 (1.4)	0.002	0.01
fenthion	88.8 (6.5)	92.2 (4.0)	84.6 (4.7)	0.002	0.01	87.5 (0.6)	91.0 (2.8)	84.8 (1.4)	0.002	0.01
diethofencarb	72.3 (2.1)	90.1 (3.5)	80.2 (6.2)	0.002	0.01	87.9 (3.3)	81.0 (2.3)	77.9 (0.7)	0.002	0.01
chlorpyrifos	82.9 (5.7)	99.8 (5.3)	85.5 (2.3)	0.002	0.01	78.8 (3.6)	77.9 (3.8)	73.0 (0.3)	0.002	0.01
triadimefon	99.4 (0.9)	90.5 (3.5)	94.9 (4.2)	0.002	0.01	85.0 (2.4)	109.1 (1.0)	93.4 (1.9)	0.002	0.01
isocarbophos	73.7 (4.6)	77.5 (4.1)	89.2 (3.2)	0.002	0.01	89.8 (5.5)	98.5 (1.3)	98.6 (0.9)	0.004	0.01
cyprodinil	104.0 (2.0)	86.9 (3.3)	97.7 (1.5)	0.002	0.01	72.0 (1.5)	73.3 (4.4)	83.2 (1.8)	0.002	0.01
metazachlor	93.9 (5.2)	91.9 (4.4)	82.2 (6.7)	0.002	0.01	106.0 (2.2)	105.1 (4.5)	87.5 (0.9)	0.002	0.01
pendimethalin	84.1 (1.9)	80.8 (4.6)	72.8 (3.9)	0.002	0.01	86.8 (8.3)	78.6 (3.6)	74.7 (1.1)	0.002	0.01
chlorfenvinphos	103.1 (2.0)	81.8 (1.2)	100.0 (2.6)	0.002	0.01	104.2 (3.5)	100.0 (6.5)	108.8 (1.4)	0.002	0.01
fipronil	95.6 (5.2)	94.6 (7.3)	87.1 (5.2)	0.002	0.01	98.7 (8.2)	87.3 (4.4)	97.4 (0.1)	0.002	0.01
procymidone	105.1 (2.1)	98.7 (2.3)	86.8 (3.2)	0.002	0.01	92.2 (2.0)	94.1 (4.2)	85.1 (2.1)	0.002	0.01
vinclozolin	96.2 (1.4)	79.5 (3.7)	82.7 (3.4)	0.002	0.01	85.0 (2.8)	99.9 (3.5)	90.3 (1.2)	0.002	0.01
methidathion	76.7 (1.6)	108.4 (2.9)	92.8 (4.1)	0.002	0.01	102.5 (5.6)	98.2 (4.9)	104.8 (1.5)	0.002	0.01
butachlor	89.3 (6.7)	90.8 (1.9)	84.2 (4.7)	0.002	0.01	101.2 (2.4)	99.0 (3.7)	92.7 (0.7)	0.002	0.01
flutriafol	93.8 (0.9)	85.2 (0.5)	84.2 (2.3)	0.002	0.01	94.0 (2.3)	92.4 (2.1)	94.5 (2.5)	0.002	0.01
carbaryl	-	80.7 (4.0)	97.4 (4.0)	0.03	0.05	86.6 (2.4)	85.9 (3.9)	70.2 (1.3)	0.005	0.01
napropamide	92.5 (2.0)	89.9 (1.9)	96.0 (0.9)	0.002	0.01	84.3 (2.1)	98.0 (0.8)	93.6 (0.6)	0.002	0.01
hexaconazole	95.4 (0.6)	96.0 (2.8)	97.3 (1.7)	0.002	0.01	90.1 (0.9)	90.2 (1.4)	82.7 (1.0)	0.002	0.01
profenofos	94.1 (2.5)	83.9 (3.0)	100.7 (5.6)	0.002	0.01	100.8 (3.9)	91.7 (1.8)	100.8 (2.5)	0.002	0.01
oxadiazon	97.2 (3.5)	92.6 (1.8)	84.1 (5.3)	0.002	0.01	96.3 (1.6)	97.3 (3.6)	87.4 (1.0)	0.002	0.01
iprovalicarb II	-	96.2 (4.3)	98.0 (4.1)	0.01	0.05	-	96.4 (1.9)	91.0 (2.5)	0.02	0.05
carboxin	106.1 (2.7)	90.6 (4.9)	89.5 (1.2)	0.002	0.01	85.8 (3.5)	89.7 (3.8)	93.7 (0.9)	0.002	0.01
oxyfluorfen	87.0 (0.8)	101.8 (4.8)	84.7 (2.5)	0.002	0.01	83.5 (3.0)	88.4 (7.7)	84.9 (1.4)	0.002	0.01
flusilazole	-	92.4 (5.3)	90.9 (4.2)	0.01	0.05	86.8 (1.6)	85.1 (1.0)	85.3 (3.1)	0.002	0.01
kresoxim-methyl	83.6 (2.9)	84.9 (0.8)	81.4 (4.2)	0.002	0.01	101.0 (1.4)	95.5 (2.3)	88.4 (1.7)	0.002	0.01
metalaxyl	79.8 (2.3)	78.0 (2.2)	84.8 (4.3)	0.002	0.01	92.0 (6.3)	90.5 (3.4)	88.8 (0.8)	0.002	0.01
diniconazole	105.4 (2.4)	106.0 (0.8)	99.0 (5.5)	0.002	0.01	96.4 (1.8)	94.7 (2.8)	92.7 (1.7)	0.002	0.01
triazophos	-	86.0 (1.8)	95.5 (3.2)	0.01	0.05	100.8 (5.6)	93.8 (2.4)	94.5 (2.4)	0.002	0.01
propiconazole Ⅰ	95.0 (5.1)	89.8 (0.5)	99.4 (4.5)	0.002	0.01	97.0 (2.1)	90.0 (1.1)	94.2 (2.7)	0.002	0.01
propiconazole Ⅱ	96.5 (2.5)	92.1 (1.3)	97.1 (0.2)	0.002	0.01	91.7 (4.0)	89.6 (2.1)	94.9 (3.0)	0.002	0.01
propyzamide	92.8 (3.4)	85.6 (2.3)	101.9 (2.6)	0.002	0.01	98.3 (2.1)	91.3 (2.2)	94.2 (2.9)	0.002	0.01
diclofop-methyl	99.2 (2.3)	89.6 (3.1)	96.1 (4.5)	0.002	0.01	93.3 (4.6)	92.6 (5.3)	81.2 (1.3)	0.002	0.01
epoxiconazole	-	72.4 (3.7)	76.9 (2.8)	0.02	0.05	-	75.7 (6.0)	77.1 (0.4)	0.02	0.05
iprodione	83.4 (4.2)	93.8 (2.1)	103.8 (7.5)	0.002	0.01	104.1 (1.9)	77.8 (3.5)	79.9 (2.2)	0.002	0.01
cypermethrin-Ⅰ	87.1 (0.8)	89.6 (2.3)	78.6 (4.0)	0.002	0.01	94.7 (5.0)	102.5 (3.4)	93.3 (0.8)	0.002	0.01
bifenthrin	85.0 (3.1)	89.9 (4.5)	85.7 (1.6)	0.002	0.01	98.9 (3.4)	85.6 (2.3)	84.9 (1.0)	0.002	0.01
bifenox	102.6 (2.3)	90.7 (6.8)	87.9 (4.5)	0.002	0.01	97.7 (1.6)	104.3 (1.2)	103.4 (3.8)	0.002	0.01
pyriproxyfen	83.5 (0.7)	85.1 (4.2)	83.0 (3.0)	0.002	0.01	86.3 (0.6)	79.0 (2.2)	73.9 (0.4)	0.002	0.01
cypermethrin Ⅱ	-	91.7 (2.9)	93.4 (1.6)	0.01	0.05	91.0 (2.8)	100.6 (2.0)	83.7 (0.8)	0.002	0.01
beta-cypermethrin	-	91.7 (2.9)	93.4 (1.6)	0.01	0.05	91.0 (2.8)	100.6 (2.0)	83.7 (0.8)	0.002	0.01
cypermethrin Ⅲ	-	103.6 (4.5)	94.2 (3.9)	0.01	0.05	98.5 (5.5)	105.8 (2.5)	105.4 (1.9)	0.002	0.01
permethrin Ⅰ	-	94.7 (2.3)	97.2 (3.2)	0.01	0.05	-	104.2 (0.8)	104.9 (2.5)	0.02	0.05
pyridaben	98.4 (5.5)	97.9 (2.2)	96.5 (4.9)	0.002	0.01	103.4 (2.9)	83.3 (1.3)	97.2 (0.8)	0.002	0.01
permethrin Ⅱ	83.7 (1.7)	83.9 (2.1)	98.1 (4.6)	0.003	0.01	95.7 (2.0)	93.2 (1.5)	96.0 (2.7)	0.002	0.01
cypermethrin Ⅳ	-	94.7 (3.0)	95.7 (2.3)	0.01	0.05	-	101.7 (1.1)	92.9 (1.9)	0.01	0.05
difenoconazole	94.2 (4.3)	93.7 (2.6)	95.1 (3.5)	0.002	0.01	94.6 (2.8)	89.9 (1.0)	93.1 (2.2)	0.002	0.01
azoxystrobin	-	98.2 (6.1)	95.2 (1.7)	0.01	0.05	-	91.2.0 (4.1)	103.6 (1.3)	0.01	0.05
deltamethrin Ⅰ	-	95.8 (3.3)	91.6 (4.6)	0.01	0.05	96.4 (3.2)	101.3 (0.7)	98.8 (1.8)	0.002	0.01
deltamethrin Ⅱ	-	96.1 (2.7)	95.2 (3.5)	0.01	0.05	98.3 (1.1)	97.9 (3.6)	84.8 (2.1)	0.002	0.01

**Table 4 foods-10-02731-t004:** Comparison of the proposed method with other QuEChERS methods.

Method	Detecting Instrument	Recoveries (%)	RSD (%)	LOQ (µg/kg)	Number of Pesticides	Cleanup Time Cost per Sample (Min)
m-PFC Method	GC-MS/MS	70.2–108.8	≤9.2	10–50	72	≤2
Payá et al. [45]	GC-MS/MS, LC-MS/MS	60–127	1.2–16.7	10	42	≥10
Romero-González et al. [46]	UHPLC-MS/MS	70–120	≤24	10	90	≥10
Martínez et al. [47]	LC/MS/MS	73–87	2-16	10	9	≥10
Santana-Mayor et al. [17]	UHPLC-(Q-ToF)-MS/MS or GC-QqQ-MS/MS	75–100	5–20	2.6–21.39	173	≥10
Bernardi et al. [48]	UHPLC-(HR)MS/MS	70–120	≤20	10	90	≥10
Kosma et al. [49]	UHPLC-Orbitrap-MS	71.2–125	≤11	2.5–73		≥10
Schusterova et al. [7]	UHPLC-(HR)MS/MS	70–120	1–20	1	367	≥10

**Table 5 foods-10-02731-t005:** The detected concentrations of pesticides in the real samples from supermarkets in Beijing.

Pesticide	X ± SD ^a,n^						
	Red Wine-03	Red Wine-04	Red Wine-15	Red Wine-22	Red Wine-27	Red Wine-46	White Wine-03
difenoconazole	ND	0.010 ± 0.001	ND	ND	0.016 ± 0.003	ND	ND
pyridaben	ND	ND	0.012 ± 0.003	0.019 ± 0.004	ND	0.012 ± 0.003	ND
carbosulfan	ND	ND	ND	ND	ND	0.026 ± 0.005	0.054 ± 0.011
pyrimethanil	ND	ND	0.031 ± 0.006	ND	ND	ND	ND
propyzamide	ND	ND	0.035 ± 0.007	ND	ND	ND	ND
simazine	0.014 ± 0.003	0.013 ± 0.002	0.042 ± 0.009	ND	ND	ND	ND
atrazine	0.015 ± 0.004	0.016 ± 0.003	0.012 ± 0.002	ND	ND	ND	ND

^a^ The average concentration (X) and standard deviation (SD) of each compound were calculated considering the pesticide residues below the limit of quantification (LOQ) for the method as non-detected(ND); ^n^ the number of 3 times each sample had been analyzed.

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
