# Peer review of "Utilizing a Rapid Multi-Plug Filtration Cleanup Method for 72 Pesticide Residues in Grape Wines Followed by Detection with Gas Chromatography Tandem Mass Spectrometry"

_foods, 2021, doi:10.3390/foods10112731_

Round 1
Reviewer 1 Report
Authors describe an application, for pesticides determination in wines, of their first developed modified QuEChERS method, based on the use of MWCNTs and m-PFC. 72 pesticides were monitored in wines by gas chromatography tandem mass spectrometry and an attempt of method validation was carried out.
The paper does not show novelties from an analytical point of view since it is a simple application of a well-documented method, also by the same Authors, to a different food matrix: the wine. Moreover, the literature is plenty of papers dealing with QuEChERS cleanup of wines for the detection of pesticides. Author quote in the introduction a few number (refs. 38-44) and the less important published papers on QuEChERS for wine sample cleanup. Among these, references 42 and 43 do not deal with QuEChERS, while 44 is about mycotoxins in wine and not pesticides.
Authors do not take into account the most important papers on this topic, in particular those published recently: Analytical and Bioanalytical Chemistry 389, 2007, 1697; Journal of Chromatography A 1218, 2011, 1477; Journal of AOAC International 99, 2016, 5; Analytical Methods 12, 2020, 2682 Foods 9, 2020, article number 1555; Foods 10, 2021, article number 307; Food Chemistry 36030, 2021, article number 130008.
These papers must be quoted and discussed properly. Moreover, Authors at least should demonstrate that their approach produces better results even in terms of performance parameters with respect to such already published papers, which deal with QuEChERS cleanup for the detection of pesticides in wine. In particular, a punctual comparison is needed with those reporting a simultaneous detection of a high number of analytes (quite more than 72).
Concerning the method validation, selectivity, one of the most important validation parameters, has not been evaluated. Authors should refer to the right EU guidelines for the validation (i.e. SANTE/2020/12830, see section 3.6 for selectivity/specificity, and EC 657/2002) and not to old (i.e. SANCO/10684/2009) or incorrect (ref. 55 does not deal with EU guidelines) documents.
Author Response
Dear reviwer:
Thanks for your good suggestions.
We have complished the responses according to your suggestions. Please see the attachments.
Best wish.
shaowen Liu

Reviewer 2 Report
Congratulations for the novel aspect of this application. In spite of the manuscript has a good potential, the data and the results were not well presented and valorised. The english language is very poor. I recommend major revisions or re-submit the manuscript after a big revision.
15: please specify the acronym MWCNT.
16: please specify the acronym m-PFC.
52: please add “They” at the beginning of the sentence
68: please replace “was” with “were”
70: please replace “it was” with “they were”
72÷80: please rephrase the part of the m-PFC description and write it shorter
81: please replace “development” with “develop”
131: please replace “two” with “three”
132: please convert the values and the units in mg/kg (according to Reg CE 396/2005)
152: please calculate the validation parameters (precision and accuracy) with the three investigated spiking levels.
156: it is necessary the respect of the abundance ion ratio to define the LOQ. You only consider one taget ion. Please verify this condition (±30%) to confirm your declared LOQ.
160: please replace “presences” with “presence”
212: please add the reference of the guidelines instead of a paper (reference n.55)
216: please replace “valuate” with “evaluate”
225: please follow the last revision of the SANTE guideline (SANTE/12682/2019)
Figure 2 and the other chromatograms are not cited in the manuscript. Please use the english language in x-axis and add the measure unit in the y-axis.
Figure 4 is not clear. For example you could overlap the traces to emphasize the effect of the pulling and pushing of the syringe. It is important to show the intensity values for each condition.
Please locate the Table 2 in supplementary materials.
Please correct the word Whit in the Table 3.
Please insert a table with the detected concentrations in the real samples analysis.
Author Response

(The authors gave the same response as above.)

Round 2
Reviewer 1 Report
There are no further suggestions or criticisms.
Author Response
Thank you for your positive and contructive comment and suggestions on our manuscript.
Reviewer 2 Report
Dear authors,
the manuscript has been improved, but in my opinion it is necessary the last minor revision:
Figure 4 is not well presented. It is important to show the intensity values for each condition.
I suggest to create a chromatogram for each experiment
Author Response
Thank you for positive and contructive comments and suggestions on our manuscript.
We have created a chromatogram for each experiment and marked the initial value for each cleanup sample accroding your suggestions.Please see the attachment.
